# Trident: A Universal Framework for Fine-Grained and Class-Incremental Unknown Traffic Detection

## ABSTRACT

To detect unknown attack traffic, anomaly-based network intrusion detection systems (NIDSs) are widely used in Internet infrastructure. However, the security communities realize some limitations when they put most existing proposals into practice. The challenges are mainly concerned with (i) fine-grained emerging attack detection and (ii) incremental updates/adaptations. To tackle these problems, we propose to decouple the need for model capabilities by transforming known/new class identification issues into multiple independent one-class learning tasks. Based on the above core ideas, we develop Trident, a universal framework for fine-grained unknown encrypted traffic detection. It consists of three main modules, *i.e.,* tSieve, tScissors, and tMagnifier are used for profiling traffic, determining outlier thresholds, and clustering respectively, each of which supports custom configuration. Using four popular datasets of network traces, we show that Trident significantly outperforms 16 state-of-the-art (SOTA) methods. Furthermore, a series of experiments (concept drift, overhead/parameter evaluation) demonstrate the stability, scalability, and practicality for Trident.

## CCS CONCEPTS

• **Security and privacy** → **Network security**; • **Information systems** → **Traffic analysis**.

## KEYWORDS

Fine-grained unknown traffic detection, class-incremental learning

## 1 INTRODUCTION

Traffic analysis is an important mechanism for security investigation, such as network intrusion detection systems (NIDSs), malware identification, *etc.* With encrypted traffic transmission becoming ubiquitous in practice, the proposed approaches gradually evolve from signature-based [13, 29] to machine-learning-based (ML-based) detection in the traffic analysis landscape [18]. These ML-based schemes aim to characterize traffic patterns with packet fields or sequence features since the transmission content is encrypted, and they are able to detect unknown attacks. For instance, anomaly-based solutions construct profiles of benign traffic to discover unforeseen attacks that deviate from legitimate samples [13, 14, 31, 38, 41, 65]. Thus, anomaly detection becomes an indispensable step for security in the real world. However, academic communities and industrial practitioners reveal a series of limitations when they put most existing anomaly-based proposals into practice [10, 23, 31, 52]. By summarizing those issues, we recognize the following two main challenges.

(i) *Fine-grained unknown attack detection.* The anomaly-based methods can identify unknown attacks, while previous proposals are usually binary classification models[1] [7, 38, 41]. That is to say,

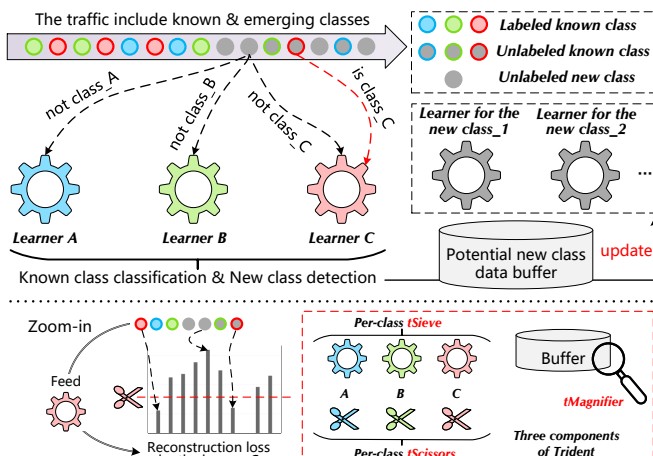

**Figure 1: The workflow of Trident and three components.**

they could only infer whether the sample is "benign" or "abnormal", but cannot recognize that the anomaly is "unknown attack 1", "unknown attack 2", or "unknown attack 3", *etc.* Yet these fine-grained labels are the key for defenders to deploy effective countermeasures against the attacks [10, 31]. For example, the victims can count the per-flow protocol flag to mitigate flood-based DDoS [37, 69]. Also, they can defend the reflection-based attacks by source verification [46]. Therefore, the binary classification anomaly detection leads to a *semantic gap* between the model identification results and the actionable reports for network operators [52]. If the proposed scheme can automatically distinguish different unknown attack classes based on the network traffic characterization, it could facilitate understanding attack details and implementing corresponding countermeasures. Consequently, the first challenge is to detect the unknown (and known as well) attack in a fine-grained manner.

(ii) *Incremental update.* Incremental update requirements include *sample increments* and *class increments*. The former means the known-class traffic that could be ever-changing (also known as the concept drift problem [3, 7, 17, 24, 66]). In most anomaly-based detection, they advocate only using benign traffic to train the classifier, *i.e.,* "zero-positive" learning [7, 18], then those samples deviating from legitimate traffic will be considered malicious. In this way, it could appear a large number of false positives when the legitimate traffic manifested as different from priori properties. Another requirement "class increments" refers to detected emerging classes that should be incrementally updated into the model to become known classes in the follow-up. Therefore, it is non-trivial to attach new classes to the model's knowledge base without affecting the previously known classes.

In this paper, we propose a universal framework, named as Trident[2], aiming to enable process three abilities, *i.e.,* the known class

---

[1]Some existing multi-class detection methods have strong assumptions, and we summarized them in baselines (§ 7.1).

[2]In ancient Greek mythology, Trident is the weapon of Poseidon, symbolizing great power. Our proposal is designed to perform powerful detection capabilities for known/unknown traffic.

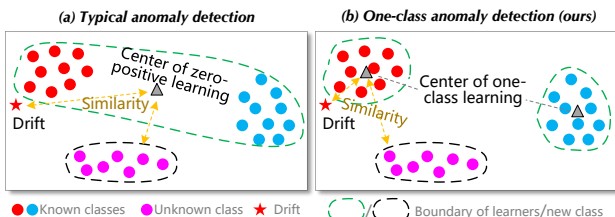

**Figure 2: Illustration of coping with concept drift.**

classification, fine-grained unknown class detection, and incremental model update (including *sample increments* and *class increments*). At the high level, Trident is designed with three tightly coupled components named tSieve, tScissors, and tMagnifier in Fig. 1 (notably, each component supports custom configuration to meet various users' requirements).

First, tSieve maintains a series of one-class learners for known classes. Each single-class learner is built upon only one class of data. Taking the AutoEncoder as an example, the well-trained tSieve will output a smaller reconstruction loss for the same class samples, while outputting a large loss for other classes. Then, tScissors is responsible for deploying the outlier detection algorithm to automatically determine the threshold for each one-class learner, which decides whether per-sample is accepted or rejected by tSieve. If a sample is accepted by at least one learner in tSieve, it is considered a known class and tagged with the most similar label. Otherwise, those samples rejected by all learners will enter the buffer pool to further assign fine-grained new class labels through the tMagnifier clustering process.

The main benefit of such a framework is that it decouples the requirements of model capabilities, and each learner only needs to focus on one class of samples. In this way, when an emerging class arrives, only a new learner needs to be added without changing the existing learner. Meanwhile, model capability decoupling can mitigate the impact of concept drift to some extent. As shown in Fig. 2 (a), typical zero-positive learning fits a wide variety of legitimate flows, forming a generalized sample center (marked as a grey triangle). This makes it difficult to distinguish drifts of benign samples from unknown attacks, since their similarity to the center is comparable. While the one-class learning of Trident is essentially a divide-and-conquer strategy, it has more distinguishable[3] for drift samples and unknown classes in Fig. 2 (b). Furthermore, if there is concept drift in any class, just incrementally update its learner and outlier threshold based on drift samples to complete the adaptation, *i.e.,* completing *sample increments*.

In summary, this paper makes three key contributions.

- We carefully examine the problems of current anomaly-based NIDSs in the real world and summarize them as two key challenges. To facilitate addressing the above issues, we propose Trident, a universal framework for fine-grained traffic detection.
- We design three tightly coupled components (tSieve, tScissors, tMagnifier) for Trident, thereby realizing the known class classification, fine-grained unknown class detection,

---

[3]It refers to that for a one-class learner, the unknown-class instances tend further away from the single-class sample center than the drift sample.

and incremental model update simultaneously. Notably, they all support the customized configuration.
- Through experiments, we demonstrate that Trident significantly outperforms previous methods. Meanwhile, we conduct a series of additional experiments to show its superior stability, scalability, and practicality.

## 2 ASSUMPTIONS AND PROBLEM SPACE

### 2.1 Threat Model and Assumptions

**Adversary Model.** We consider unknown intrusions such as zero-day attacks that exist in real-world scenarios. In other words, strong adversaries will adopt the emerging attack strategies that are previously unseen by victims, including variants of the existing attacks or brand new ones. Therefore, it is hard to have any prior data about these unforeseen attacks. Moreover, the traffic of known types is not set in stone. Therefore, the problem of concept drift is within the scope of consideration. In addition, we mainly focus on *encrypted traffic analysis* in this paper, since the transmission content is increasingly being encrypted in existing networks, such as SSL/TLS and SSH. Concretely, we tend to characterize traffic behavior by portraying packet field distribution rather than analyzing transmission content, *e.g.,* TCP Payload.

**Assumptions.** We assume there is no prior knowledge when suffering unprecedented attacks in practice. We are also not aware of how many types of unknown attacks exist in the collected traffic samples in advance. Meanwhile, we do not assume additional collaborations from other Internet entities, such as IP blacklists provided by security vendors. In addition, multiple unknown attacks may be launched at the same time or may appear alternately. That is to say, we do not make assumptions about the distribution of attacks.

### 2.2 Problem Formulation

We provide precise definitions of two critical challenges.

**Fine-Grained Unknown Attack Detection.** Given a prior dataset $S_{tra}$, consisting of samples from benign traffic $B$ and known cyber attacks $\{A_k^1, A_k^2 \cdots A_k^n\}$, where $n$ refers to the number of known classes. And we use $S_{tra}$ as the training set to fit the model $M$. When deploying $M$ in practice, it will encounter the open-world testset include: (i) samples $S_k$ with the ground-truth labels from $\{B, A_k^1, A_k^2 \cdots A_k^n\}$; (ii) instances of emerging classes $\{A_u^1, A_u^2 \cdots A_u^m\}$, where $m$ denotes the number of unknown classes, and $m$ is unknown to us in advance. The *fine-grained unknown attack detection* refers to: $M$ can identify the specific-attack labels of test samples, *i.e.,* the sample prediction result is $B$, $A_k^1$, $A_k^2$, $A_u^1$, $A_u^2$, or others[4].

**Incremental Update.** The requirements of incremental updates include two parts: *sample increments* and *class increments*. For sample increments, it refers to the model adapting to new instances without retraining the whole architecture. This capability stems from two considerations, for one thing, the business scenario changes will cause different manifestations of legitimate traffic (*e.g.,* the traffic of

---

[4]Note that the model marks the attack with a series of fine-grained codenames (*e.g.,* "new attack 1", "new attack 2"), rather than naming each attack (*e.g.,* "Heartbleed"). We aim to automatically detect unknown attacks and distinguish their types from each other in this work, which benefits network operators to analyze and further deploy countermeasures. The specific attack names can be given by the security communities, just like we could also call "Heartbleed" as "Buffer over-read".

streaming media and chatting would be very different). For another, sample increments could also be used to adapt to the concept drift of attack (such as some attacks changing over time, making the previous attack traffic somewhat outdated).

Regarding class increments, it is actually an inevitable problem for fine-grained detection, and it is also a major difference between us and existing work. When a new type of attack $A_u^*$ is detected, we hope to add it to the knowledge base to strengthen the model's capability. In other words, the updated model can detect $A_u^*$ just like supervised classification in the future. This is a challenging task that cannot be achieved by most existing works.

## 3 TRIDENT FRAMEWORK OVERVIEW

In this section, we depict a high-level workflow of Trident, including the key building blocks of each component. Most typical schemes are either supervised learning for multi-class classification or binary classification anomaly detection. We architecture Trident as a novel framework to decouple the requirements of model capabilities. The overall process includes three parts: one-class learning, outlier determination, and fine-grained label assignment.

**One-Class Learning.** First, unlike typical models, Trident maintains a collection of one-class learners, each of which corresponds to a traffic type one-to-one. Each learner is only responsible for reconstructing the corresponding type of traffic data, and the learners are independent of each other. Therefore, when a test sample arrives, each learner outputs its reconstruction loss and decides to accept or reject this instance. The instance that cannot be accepted by any learner is considered the new class, otherwise, it will be assigned as the most similar class among all accepted learners.

**Outlier Determination.** Then, determining the outlier threshold is to help the learner decide whether to accept or reject an instance. In particular, each one-class learner has its own outlier threshold, independent of each other. The determination method is mainly based on the analysis of previous data, and the specific threshold is manually preset or obtained by probability and statistics.

**Fine-Grained Label Assignment.** The need for fine-grained labels includes both known and unknown classes. For known classes, the labels will be jointly derived from the results of multiple learners, as described above. Those samples that are not accepted by any learner refer to unknown classes and will enter the buffer. In the buffer, all unknown classes are assigned labels through a series of unsupervised strategies. Finally, samples are labeled as specific known types or fine-grained unknown classes. Noteworthy, these detected unknown classes can be used to build new one-class learners, given that Trident supports class increments.

## 4 TSIEVE: ONE-CLASS LEARNING

In this section, we provide three model architectures to perform the one-class learning task. Each model architecture corresponds to its unique feature vector extraction, but all are based on the traffic session with the 5-tuple index, i.e., {Source IP, Source Port, Destination IP, Destination Port, Protocol}.

### 4.1 AutoEncoder

**Feature Extraction.** For AutoEncoder, Table 4 shows the feature generated for each bi-directional flow. It characterizes the traffic in

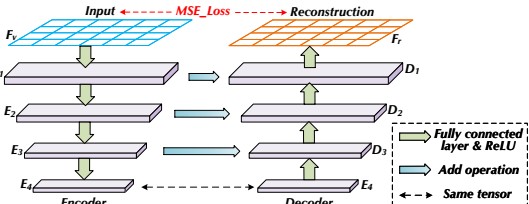

**Figure 3: The AutoEncoder architecture in Trident.**

terms of temporal, volumetric, and header-field distributions. Specifically, including (i) the protocol coding with one dimension. (ii) The count of a series of flags from layers 1-4 (such as IP Fragment, TCP Flags) in three situations (forward, backward, and bidirectional), which is 33 dimensions. (iii) The 72-dimensional statistical results (i.e., max, min, mean, and std) for several transport-functional fields (i.e., TTL, window size) in three situations about direction. In general, all features are either int or float types.

**Model Architecture.** A vanilla design is directly using the typical AutoEncoder (AE) as the one-class learner. However, deep layers in this typical architecture may lose some feature information. To this end, we add the tensor from the encoder to the corresponding layer in the decoder, so as to preserve more semantic information in different tiers. The model architecture is shown in Fig. 3, which is somewhat similar to U-Net [45] (a convolutional network). It contains an encoder and a decoder, which are symmetrical to each other. For example, in the encoder, the $d$-dimension input is concatenated with four fully connected layers (with the ReLU activation), in which the corresponding feature vectors are $F_v \rightarrow E_1 \rightarrow E_2 \rightarrow E_3 \rightarrow E_4$ and the dimensions refer $d \rightarrow 256 \rightarrow 128 \rightarrow 64 \rightarrow 32$. Notably, the decoder part will cascade the intermediate output of the encoder for addition operation (denote as the blue arrow in Fig. 3). For instance, $D_2$ can be calculated by $D_2 = ReLU(Linear(D_3 + E_3))$. Such an addition operation preserves richer features at each level to alleviate information loss during reconstruction. Meanwhile, we employ mean squared error (MSE) as the loss function since the per-learner is essentially performing the data reconstruction. Empirically, a well-trained learner will output a smaller MSE loss for same-class samples, while data of different classes will lead to a larger loss.

### 4.2 Recurrent Neural Network

**Features Extraction.** For input to the Recurrent Neural Network (RNN) model, we extract the packet field from the raw traffic and expand the feature according to time to form sequences. Subsequently, all feature vectors are pruned or padded to $T$ time steps. Per-packet feature vector consists of packet length, direction, the interval of arrival time, and values of fields in the packet header, e.g., TTL (Time To Live), TCP flags, TCP window size, and SSL/TLS

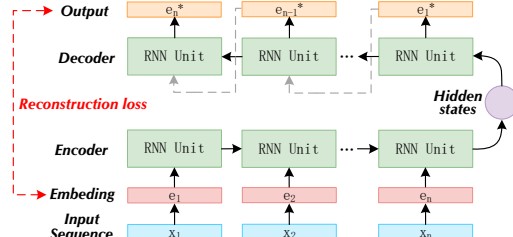

**Figure 4: The RNN architecture in Trident.**

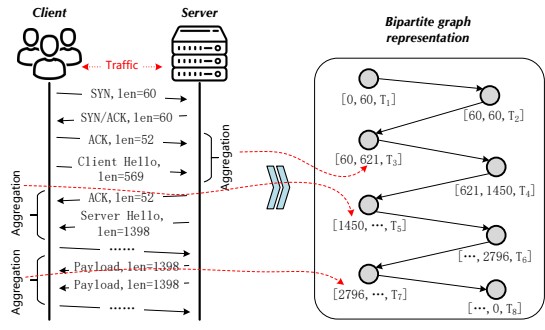

**Figure 5: The bipartite graph representation.**

fields. Therefore, a sample can be denoted as $x = [L_1, L_2, ..., L_T]$, where $T$ is the time steps and $L_i \in \mathbb{R}^d$ is the $d$-dimensional packet feature at time step $i$. Such feature extraction is similar to most sequence-based methods, except that we focus on more fields (such as SSL/TLS) than just packet length and timestamp [34, 70].

**Model Architecture.** To accomplish data reconstruction as a one-class learner, we consider a representative Seq2Seq model [55]. The model architecture (more details in § A.2) is shown in Fig. 4, it consists of an encoder and a decoder of equal length. We cascade an *embedding layer* between the input and the RNN unit to reduce the impact of feature values, given different feature fields may have different value ranges. Formulaically, the $x_i$ in each time step will go through the embedding layer to obtain $e_i$ and then be fed to the encoder's RNN unit $U_{RNN}^E$. The hidden state output of the RNN unit is passed at different time steps. Afterward, the hidden state for the last time step of the encoder will be fed to the decoder as the initial state. The input to the decoder is the output of the previous time step (except for the first time step), with the hidden state passed. We collect the outputs $e_i^*$ of decoder units $U_{RNN}^D$ at all time steps. The reconstruction loss is still calculated with MSE, *i.e.*,

$$\mathcal{L} = \sum_{i=1}^{n} \sqrt{(e_i{}^2) - (e_i{}^*)^2} \quad (1)$$

### 4.3 Graph Model

**Features Extraction.** The third model is a graph-based approach. To this end, we develop a novel session graph construction to extract features. As Fig. 5 displays, it considers the bidirectional traffic interaction between the client and server. We aggregate those adjacent packets in the same direction, forming an edge that includes the aggregated bytes. Based on such a process, the nodes are distributed on either the client or server. In the end, a bipartite graph can be obtained, which is characterized in that the points of the client and the client will not be connected, and the same applies to the server. An example of packet aggregation can be found at the bottom of Fig. 5, two TCP payload packets with a length of 1,398 are aggregated to form an edge with 2,796 bytes. Thus, the feature vector of a node includes three elements, *i.e.*, [received bytes, sent bytes, duration]. The advantage of such data representation is that it has robustness, *e.g.*, it can alleviate the influence of changes from the Content Delivery Network (CDN) and Maximum Transmission Unit (MTU) on traffic analysis.

**Model Architecture.** For the above bipartite graph, we employ the U-Net-like GNN architecture [45], which is depicted in Fig. 6 (more details in § A.3). Among the model input, the traffic graph

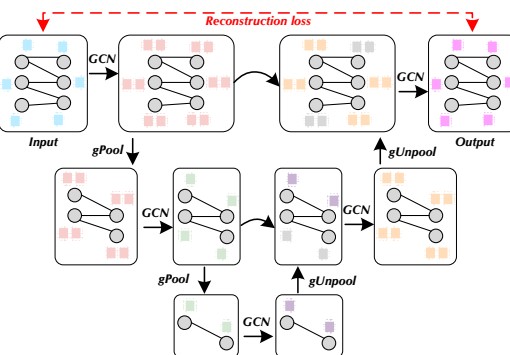

**Figure 6: The GNN architecture in Trident.**

$\mathbb{G}$ can be represented by the adjacency matrix $A \in \mathbb{R}^{N \times N}$ and the feature matrix $X \in \mathbb{R}^{N \times C}$, where $C$ and $N$ represent the nodes' feature dimension and number, respectively. Also, each row vector $X_i = [x_i^0, x_i^1, \cdots, x_i^C]$ in the feature matrix $X$ denotes the feature vector of node $i$ in the graph. During the encoder stage, the node feature vectors are first transformed into low-dimensional representations using a GCN layer. After that, two encoder blocks are performed, each of which contains a gPool layer [15] and a GCN layer. Correspondingly, there are two decoder blocks in the decoder part, and each block consists of a gUnpool layer [15] and a GCN layer. For blocks at the same level, the encoder block uses a skip connection to fuse the low-level spatial features from the encoder block. Finally, the output feature vectors $X_i' \in X'$ of nodes in the last layer are network embedding, which can be used for computing the reconstruction loss $\mathcal{L}$.

$$\mathcal{L} = \sum_{i=1}^{N} \sum_{j=1}^{C} \sqrt{(x_{ij}')^2 - (x_{ij})^2} \quad (2)$$

### 4.4 Incremental Update

**Sample-Level Incremental Update.** To adapt to new samples, we don't need to retrain the model, and only perform model generalization at the sample level. A strawman design might be to combine old data with new instances for model updates. The potential problem with this solution is that the scale of the data will continue to expand over time, resulting in inefficiency. Therefore, we could consider selecting some representative historical data. Recall the one-class learner reconstructs samples and computes an outlier threshold for the reconstruction loss. In order to keep the distribution of the reconstruction loss list as unchanged as possible before and after sampling, we sort the reconstruction loss list and sample them at intervals. For example, Fig. 7 (b)-(c) display the sampling strategy when setting sample rate $r$=0.5, 0.2 respectively, where red columns represent sampled instances. In practice, the sampling rate can be set according to user requirements. These sampled historical data are used together with new data to incrementally update one-class learner parameters.

**Class-Level Incremental Update.** Since each one-class learner is independent of each other, the update operation can be conveniently performed. Using each new class to build the corresponding one-class learner and populate the learner list and threshold list, as shown in lines 1~7 in Algorithm 2. After updating, the new class can then be viewed as the known class. Overall, due to learners corresponding to classes one by one, building learners for new classes

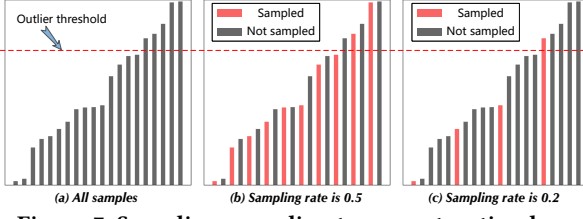

**Figure 7: Sampling according to reconstruction loss.**

is incremental-manner. Meanwhile, the special advantage of this framework for model construction is that it can be run in parallel, easy to horizontal expansion, and reduce overhead, *e.g.*, distributed to deploy those one-class learners on different computing nodes.

## 5 TSCISSORS: OUTLIER IDENTIFICATION

In this section, we develop some outlier identification schemes such as preset method and extreme value theory method.

**Preset Method.** A vanilla design is to preset the outlier ratio based on expert experience/knowledge. Given the output list $L_o$ by a learner of length $N_o$, first sort it in ascending order to get $L'_o$. Consider a preset ratio $R_p$, which refers to treating the last $int(N_o \times R_p)$ data in $L'_o$ as outliers. In other words, the threshold is the outlier boundary, *i.e.*, $L'_o[int(N_o \times R_p)]$. Actually, some similar work involving outlier detection methods are preset methods. For example, in the adversarial example detection landscape, many works construct a series of metrics to score inputs and use preset outlier thresholds to distinguish normal instances and adversarial examples [39, 64, 67]. The advantage of the preset method is that it is convenient and prone to implement, but the limitation is that it needs to be manually set and sometimes lacks flexibility.

**Extreme Value Theory Method.** To solve this problem, we also leverage the Extreme Value Theory (EVT) [28] to estimate the outlier bounds. The original intention of this theory by statisticians is to study the possibility of extreme events. An elegant property of EVT is that the distribution of the extreme values is not dependent on the distribution of the data. In other words, these extreme events have the same kind of distribution, regardless of the original one, such as Frechet, Gamma, and Uniform distributions [51]. For specific details that utilize EVT to estimate thresholds, refer to § B.

## 6 TMAGNIFIER: LABEL ASSIGNMENT

We elaborate here on the process for fine-grained label assignment.
**Known Class Classification.** Algorithm 1 describes the procedure for label assignment. Consider a trained one-class learner list $L_p$ and its threshold list $L_t$, each test sample will traverse each learner to obtain reconstruction loss. If the reconstruction loss is less than the threshold, the class of this learner will be considered as a potential candidate (in line 8). Naturally, the instance that cannot match any of the learners is regarded as a new class (line 10). On the contrary, there is at least one learner that can cover the sample, *i.e.*, $len(M_c) \geq 1$, the class of the learner with the smallest reconstruction loss will be used as the prediction result (lines 12~13).
**Unknown Class Identification.** For new-class samples in the buffer pool $\mathcal{B}$, we leverage the clustering algorithm to perform fine-grained identification. Since we cannot know in advance how many unknown classes there are, a scheme such as K-means [19] that

---

**Algorithm 1** Label Assignment

**Require:** The test dataset $\mathcal{S}$, the known-class one-class learner list $L_p$, and corresponding threshold list $L_t$, buffer $\mathcal{B}$
**Ensure:** $y$ - class label for each $x \in \mathcal{S}$
1: Initialize prediction list $Result = [\ ]$, temporary lists $R_k$ and $R_u$
2: **for** $x \in \mathcal{S}$ **do**
3:     Initialize reconstruction loss list $L_r = [\ ]$
4:     **for** $p_i \in L_p$ **do**
5:         Calculate reconstruction loss $l \leftarrow p_i(x)$
6:         Apeend $l$ to $L_r$
7:     **end for**
8:     Candidate $M_c = \text{array}(L_r)[\text{array}(L_r) < \text{array}(L_t)]$
9:     **if** $len(M_c) == 0$ **then**
10:         $\mathcal{B} \leftarrow \mathcal{B} \cup x$
11:     **else**
12:         Select $class_j$ s.t. $L_r[class_j] == min(M_c)$
13:         Apeend $class_j$ label to $R_k$
14:     **end if**
15: **end for**
16: Fine-grained new class results $R_u \leftarrow Clustering(\mathcal{B})$
17: $Result = R_k \cup R_u$
18: **return** $Result$

---

**Figure 8: Unknown class identification.**

needs to set the number of clusters is not suitable. DBSCAN [9] is an optional solution. However, DBSCAN also needs to set parameters, such as *eps* or *min_samples*. Therefore, we intend to cluster with various parameters and integrate the results to achieve relatively stable results. The process of ensemble clustering ($Num$ clustering models) is as follows. Considering the $\mathcal{B}$ sample set, we construct an adjacency matrix $A_c$ to record the results of $Num$ clustering algorithms. For example, if $A_c[i, j] = \Delta$, it refers to $\Delta$ clustering algorithms to cluster samples $x_i$ and $x_j$ into one cluster, where $x_i, x_j \in \mathcal{B}$ and $\Delta \leq Num$. As shown in Fig. 8, we need to generate the integrated results of the ensemble clustering according to $A_c$. The key is to determine a cut point to filter out the smaller values in the matrix, and thereby find the connected subcluster. Therefore, we can traverse the cut points from 0 to $Num$ and calculate the silhouette coefficient [47], then select the cut point when the silhouette coefficient is the largest.

## 7 EVALUATION

We comprehensively evaluate Trident, with code available online (anonymous repository https://github.com/WWW-rep/Trident/).

### 7.1 Experiment Setup

**Datasets.** A series of public datasets are used for evaluation as follows. **(i) USTC-TFC2016** dataset [57] includes network traffic from two parts, *i.e.*, malware and benign. Among them, ten types of malware traffic from public websites are collected from real-world

Table 1: The supervised evaluation results.

| Dataset | USTC | | Tor | | IDS | | CrossNet | |
|---|---|---|---|---|---|---|---|---|
| Model | ACC | F1 | ACC | F1 | ACC | F1 | ACC | F1 |
| nPrint | **95.68** | 89.75 | **99.50** | 93.37 | 99.86 | 99.62 | 93.89 | 91.58 |
| FlowPic | 76.88 | 73.57 | 94.82 | 89.62 | 99.22 | 99.24 | 74.38 | 72.45 |
| FlowLens | 86.91 | 84.34 | 98.76 | 91.38 | 98.55 | 91.09 | 83.22 | 81.41 |
| ERNN | 90.18 | 88.72 | 98.10 | 96.15 | 98.50 | 97.19 | 86.64 | 85.12 |
| ET-BERT | 94.51 | 90.26 | 95.69 | 92.70 | **99.93** | **99.63** | 91.22 | 88.45 |
| Trident (AE) | 95.65 | **91.20** | 94.13 | 89.01 | 99.70 | 99.29 | 92.30 | 90.61 |
| Trident (RNN) | 92.38 | 89.64 | 99.38 | **97.02** | **99.93** | 99.57 | 91.17 | 87.20 |
| Trident (GNN) | 94.32 | 90.07 | 97.30 | 94.51 | 99.62 | 99.34 | **94.51** | **92.36** |

connections. Also, benign traffic from 10 categories of applications, *e.g.,* Facetime, Skype, Weibo, was captured by the network traffic simulator called IXIA BPS. **(ii) ISCXTor2016** dataset [27] consists of network traffic involving 8 classes, *i.e.,* Email, Chat, FTP, etc, was captured using tcpdump, including 22GB of data. **(iii) IDS2017&2018** datasets [11, 12] uses a proposed B-Profile system to profile the normal and intrusion behavior, including DoS/DDoS, Brute Force, Infiltration, etc. Also, they contain more than tens of millions of traffic instances based on a series of protocols, in which the packets are captured across several days involving various operating systems. **(iv) CrossNet2021** dataset [30] contains traffic data from 20 categories of applications such as 360, Sougou, and CSDN in two practical scenarios, *i.e.,* stable and production networks. The traffic was captured using tcpdump, including 2.5GB of data.

**Baselines.** We use 16 models involving supervised, unsupervised, and semi-supervised methods as baselines, covering state-of-the-art (SOTA) in the traffic analysis landscape. (i) Five supervised method: **nPrint** [20], **FlowPic** [49], **FlowLens** [4], and **ET-BERT** [32]. (ii) Five binary classification anomaly detection: **Kitsune** [41], **Whisper** [13], **DeepLog** [8], **HyperVision** [14], and **Diff-RF** [38]. (iii) Six multi-classification anomaly detection: **K-means** [19], **DBSCAN** [9], **FlowPrint** [56], **Cls-Anomaly** [65], **SENC** [42], and **FARE** [31]. Details of these baselines can be found in Appendix C.

**Metrics.** Two popular benchmarks are used to evaluate the performance for identifying emerging classes [31], including the clustering accuracy (*ACC*) and adjusted mutual information (*AMI*). Their upper bounds are all 1 and the larger values mean the better effect. For supervised evaluation and binary classification anomaly detection (AD) experiments, we additionally calculate the Precision (*Pre*), Recall (*Rec*), F1-score, and the Area Under Curve (*AUC*).

**Hyperparameter Settings.** The sampling rate $r = 0.5$ and we also evaluate $r$ in § 7.4. For outlier threshold determination, we use the EVT method by default, and traverse settings of the preset method to produce Receiver Operating Characteristic (ROC) curve in § 7.3. For unknown class clustering, three algorithms (K-means [19], DBSCAN [9], and DEC [60]) are used, by varying the hyper-parameters for each clustering algorithm to contribute $Num = 20$ models, and we vary $Num$ to develop ablation experiments in § 7.4.

## 7.2 Supervised Evaluation

When all classes are known, Trident can directly construct per-class learner. Therefore, we first compare supervised SOTA methods to explore the known class classification effect of Trident. The classification results for the four datasets are summarized in Table 1, including the accuracy and F1 score. In Tor and IDS datasets, Trident (RNN) presents the best performance. While Trident (AE) and

Table 2: The binary classification anomaly detection results.

| Dataset | USTC | | | | IDS | | | |
|---|---|---|---|---|---|---|---|---|
| Model | ACC | Pre | Rec | F1 | ACC | Pre | Rec | F1 |
| Kitsune | 92.45 | 92.57 | 92.23 | 92.41 | 93.07 | 93.52 | 92.49 | 93.07 |
| Whisper | 94.32 | 94.83 | 93.77 | 94.30 | 91.20 | 91.12 | 91.28 | 91.21 |
| DeepLog | 92.80 | 92.72 | 92.88 | 92.80 | 91.12 | 91.26 | 90.93 | 91.09 |
| HyperVision | 96.67 | **97.28** | 96.04 | 96.66 | 95.32 | 95.85 | 94.76 | 95.31 |
| Diff-RF | 93.55 | 94.36 | 92.72 | 93.53 | 92.25 | 92.98 | 91.43 | 92.19 |
| Trident (AE) | **97.25** | 97.26 | **97.29** | **97.27** | 95.45 | 95.09 | 94.91 | 95.00 |
| Trident (RNN) | 94.95 | 94.81 | 95.08 | 94.94 | **96.02** | **96.09** | **95.94** | **96.01** |
| Trident (GNN) | 96.45 | 96.73 | **96.21** | 96.47 | 93.55 | 93.31 | 93.76 | 93.53 |

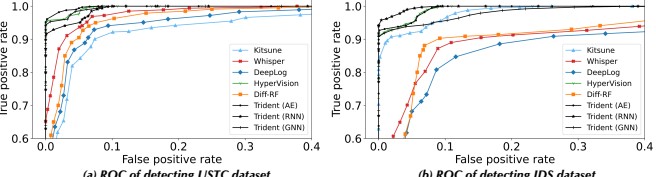

Figure 9: ROC of binary classification anomaly detection.

Trident (GNN) are prominent in USTC and CrossNet datasets, respectively. Sometimes nPrint has a slight advantage in accuracy, but the difference is only ~0.1%. Noteworthy, Trident achieves the superior F1 score, which indicates that Trident is hardly affected by data imbalance (in contrast, nPrint's F1 score drops a lot). Overall, the classification performance of Trident for known classes is competitive compared to supervised SOTA methods.

## 7.3 Binary Classification Anomaly Detection

Then, we evaluate the binary classification anomaly detection methods. Since these methods only distinguish between "benign" and "abnormal", we also change the output of Trident to "0" or "1" for a fair comparison, that is, not to distinguish the specific type of attack. Two attack-related datasets (USTC and IDS) are used to evaluate, only using their benign traffic to train for both the baseline and Trident. From Table 2, the detection results refer to Trident (AE) > HyperVision > Trident (GNN) > Trident (RNN) > Whisper > Diff-RF > DeepLog > Kitsune for USTC dataset and Trident (RNN) > Trident (AE) > HyperVision > Trident (GNN) > Kitsune > Diff-RF > Whisper > DeepLog for IDS dataset. This may be attributed to different attacks (datasets) corresponding to different optimal feature extraction and detection models. Nevertheless, Trident achieves the best binary anomaly detection performance in terms of accuracy, precision, recall, and F1 score. Moreover, we vary the detection threshold for each model to plot ROC curves in Fig. 9. According to Fig. 9 (a)-(b), we observe that Trident has fewer false positives while achieving a high true positive rate.

## 7.4 Multi-Classification Anomaly Detection

**Class Increments.** In this section, we evaluate the capability of Trident by varying $N_k:N_u$ (representing the number of known and unknown classes). We set $N_k:N_u = \{20:0, 16:4, 12:8, 8:12, 4:16, 0:20\}$ for USTC and CrossNet datasets and set $N_k:N_u = \{8:0, 6:2, 4:4, 2:6, 0:8\}$ for IDS and Tor datasets. The AMI results are shown in Fig. 10, it is clear that as the number of unknown classes increases, Trident significantly outperforms other baselines. For instance, to detect the USTC dataset in Fig. 10 (a), when all classes are known, the performance gap is not very big, *e.g.,* the AMI of Trident is slightly higher than FlowPrint by ~2%. However, with more unknown classes, the

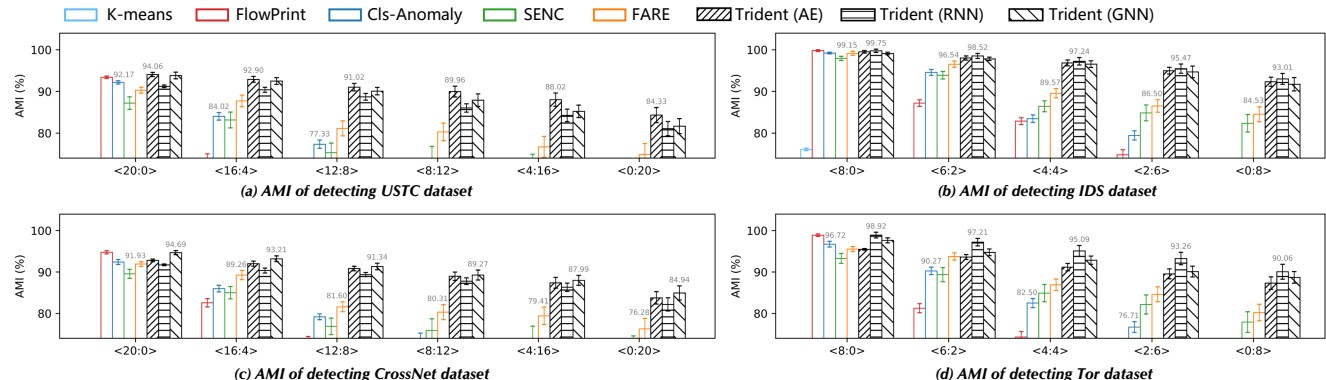

Figure 10: Evaluation in different known/unknown proportions.

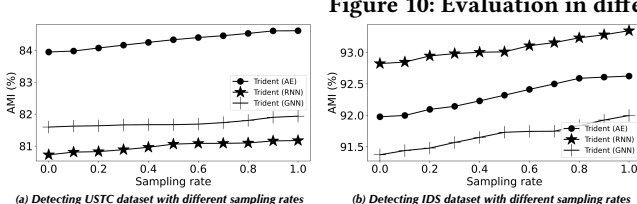

Figure 11: Multi-classification detection results of varying different sampling rates when performing sample increments.

Table 3: The ACC (%) results with different $Num$ settings.

| Cluster model $Num$ | | 5 | 10 | 20 | 50 | 100 |
|---|---|---|---|---|---|---|
| Cross. | Trident (AE) | 80.44 | 83.95 | 86.26 | 88.42 | 88.73 |
| | Trident (RNN) | 79.25 | 81.67 | 84.68 | 86.97 | 87.79 |
| | Trident (GNN) | 81.52 | 85.02 | 87.42 | 89.35 | 89.98 |
| Tor | Trident (AE) | 82.85 | 87.32 | 89.82 | 92.21 | 92.74 |
| | Trident (RNN) | 86.02 | 89.74 | 92.53 | 94.15 | 94.52 |
| | Trident (GNN) | 84.02 | 88.74 | 91.20 | 93.28 | 93.45 |

advantage of Trident becomes more obvious. When all classes are unknown, Trident outperforms other methods by ~10%. Particularly, DBSCAN does not require training, and its detection results are similar in different known/unknown proportions, *i.e.,* always less than 60% of AMI.

**Sample Increments.** With $N_k$ = 0, we vary the sampling rate $r$ when performing sample-level incremental updates, the detection results for USTC and IDS datasets are summarized in Fig. 11. It is clear that as the sampling rate increases, the AMI results gradually increase. Therefore, if space overhead allows, we could tend to select more samples for updates.

**Hyperparameter Evaluation.** We also vary the cluster model $Num$ to perform ablation experiments for $N_k$ = 0. As Table 3 shown, from $Num$ = 5 to $Num$ = 50, the accuracy increases significantly. When $Num$ = 100, the ACC change is relatively slight. Overall, Trident maintains solid performance with different parameter settings.

## 7.5 Concept Drift Evaluation

Concept drift is an unavoidable problem in anomaly detection systems, and we evaluate two scenarios *time bias* and *scenario bias.*

**Time Bias.** We conduct the time bias experiment by setting IDS2017 for training and using IDS2018 for testing, thereby exploring whether the traffic drifts over time will cause the failure of model detection. Fig. 12 (a)-(c) show the detection results for supervised, binary classification AD, and multi-classification AD models. In subfigure (a), we can see that ERNN is the best performer in the baseline because

it considers network packet loss, retransmission, and out-of-order phenomena when building the model. Other supervised baselines are greatly affected by time bias. Nevertheless, our framework shows better robustness, *e.g.,* Trident (RNN) is still able to achieve 96.71% ACC. For binary classification AD in subfigure (b), the impact of concept drift is less severe, since these are trained with only benign traffic in this setting and the impact on the model training process is limited. Trident still maintains the best performance under the binary classification AD setting. As for multi-classification AD, most of the models fail, with FARE guaranteeing an AMI of ~70% and others below 60%. In this setting, Trident realizes ~90% ACC and AMI, demonstrating its strong stability.

**Scenario Bias.** To develop the scenario bias evaluation, the Scenario A traffic is used for training and to detect Scenario B traffic from the CrossNet dataset. The two scenarios of the CrossNet dataset are collected from different network quality-of-service (QoS), such as different bandwidths and channel disturbance. The results are summarized in Fig. 12 (d)-(e). Similar to temporal bias, both the supervised and the multi-class AD models suffer to varying degrees. Also, all three model configurations of Trident outperform the baselines. Overall, the divide-and-conquer idea of Trident can effectively alleviate the impact of concept drift. In practice, we can also combine Trident with existing technologies [17] to deal with the problem of concept drift.

## 7.6 Overhead Evaluation

We measure the update time overhead for baseline algorithms and Trident, and the results are summarized in Fig. 13. All models run on the Ubuntu 18.04.2 server with Intel i7-12700K CPU, NVIDIA TITAN GPU, and 64 GB memory. Overall, nPrint, Whisper, Diff-RF, DBSCAN, K-means, and Hypervision are on one level (<1*ms*) since there are machine learning based methods. The deep learning model does have more time overhead (generally greater than 1*ms*). For example, the time overhead of ERNN, DeepLog, and Trident (RNN) is relatively similar since they are all RNN-related models. The most time-consuming model is ET-BERT because it is essentially a large language model with massive parameters.

## 8 DISCUSSION AND LIMITATIONS

**Model Selection.** We designed Trident as a universal framework that could support various model architectures to characterize traffic. This paper mainly includes three types of models: AE, RNN,

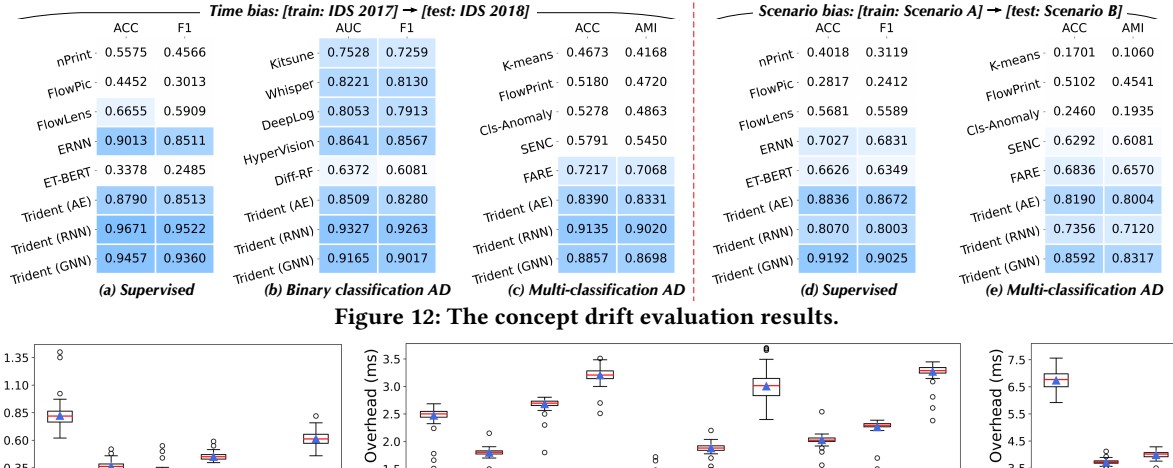

Figure 12: The concept drift evaluation results.

**Time bias: [train: IDS 2017] → [test: IDS 2018]**

(a) Supervised

| | ACC | F1 |
|---|---|---|
| nPrint | 0.5575 | 0.4566 |
| FlowPic | 0.4452 | 0.3013 |
| FlowLens | 0.6655 | 0.5909 |
| ERNN | 0.9013 | 0.8511 |
| ET-BERT | 0.3378 | 0.2485 |
| Trident (AE) | 0.8790 | 0.8513 |
| Trident (RNN) | 0.9671 | 0.9522 |
| Trident (GNN) | 0.9457 | 0.9360 |

(b) Binary classification AD

| | AUC | F1 |
|---|---|---|
| Kitsune | 0.7528 | 0.7259 |
| Whisper | 0.8221 | 0.8130 |
| DeepLog | 0.8053 | 0.7913 |
| HyperVision | 0.8641 | 0.8567 |
| Diff-RF | 0.6372 | 0.6081 |
| Trident (AE) | 0.8509 | 0.8280 |
| Trident (RNN) | 0.9327 | 0.9263 |
| Trident (GNN) | 0.9165 | 0.9017 |

(c) Multi-classification AD

| | ACC | AMI |
|---|---|---|
| K-means | 0.4673 | 0.4168 |
| FlowPrint | 0.5180 | 0.4720 |
| Cls-Anomaly | 0.5278 | 0.4863 |
| SENC | 0.5791 | 0.5450 |
| FARE | 0.7217 | 0.7068 |
| Trident (AE) | 0.8390 | 0.8331 |
| Trident (RNN) | 0.9135 | 0.9020 |
| Trident (GNN) | 0.8857 | 0.8698 |

**Scenario bias: [train: Scenario A] → [test: Scenario B]**

(d) Supervised

| | ACC | F1 |
|---|---|---|
| nPrint | 0.4018 | 0.3119 |
| FlowPic | 0.2817 | 0.2412 |
| FlowLens | 0.5681 | 0.5589 |
| ERNN | 0.7027 | 0.6831 |
| ET-BERT | 0.6626 | 0.6349 |
| Trident (AE) | 0.8836 | 0.8672 |
| Trident (RNN) | 0.8070 | 0.8003 |
| Trident (GNN) | 0.9192 | 0.9025 |

(e) Multi-classification AD

| | ACC | AMI |
|---|---|---|
| K-means | 0.1701 | 0.1060 |
| FlowPrint | 0.5102 | 0.4541 |
| Cls-Anomaly | 0.2460 | 0.1935 |
| SENC | 0.6292 | 0.6081 |
| FARE | 0.6836 | 0.6570 |
| Trident (AE) | 0.8190 | 0.8004 |
| Trident (RNN) | 0.7356 | 0.7120 |
| Trident (GNN) | 0.8592 | 0.8317 |

Figure 13: The time overhead.

and GNN. We discuss here some model selection recommendations. Based on the experimental results in § 7, we observe that RNN is superior when the number of classes is small (*e.g.,* IDS and Tor), but the performance will significantly decrease when there are more classes (refer to USTC and CrossNet datasets). When the training set and test set are identical distributions, AE is a good choice to deal with multiple categories. Compared with AE, GNN has better robustness even when there are concept drifts, this could be attributed as GNN's aggregation process alleviates the impact of changes from CDN and MTU on traffic (echo back § 4.3). Overall, users could choose desired model architectures or customize extensions/variants to cater to specific requirements.

**Attack Category Recovery.** When identifying fine-grained labels for unknown classes, it could occur to overestimate or underestimate the attack categories. A main reasons refer to he extracted protocol features have different granularities. For example, some customers may need to distinguish between different HTTP flooding and some may not. Therefore, building a customized classification scheme in the output layer according to different needs may be beneficial to promoting Trident to widespread use. We will investigate these to advance the practicality of Trident.

**Limitations and Future Works.** Our work has a few limitations. First, different customers may require various detection granularity, the future work may consider a customized scheme, *e.g.,* change the output layer of tSieve in Trident. Second, applying the automated feature extraction and model parameters tuning into Trident will lead in a good direction. Third, to provide customers with more reliable protection, the powerful adversary using a combination of multiple attacks needs to be further studied. Finally, as part of future work, we would explore which components could run in parallel to maximize efficiency.

## 9 RELATED WORK

Besides SOTA baselines in § 7.1, we list briefly some related work.

**NIDS with Known Attacks Classification.** To classify known attacks, some works [2, 35, 48, 49, 62, 63] design NIDSs based on statistical features by supervised learning methods [21, 43, 53, 68, 71], *e.g.,* random forests, deep neural networks. Some other arts utilize Markov [26, 33, 50] or recurrent neural networks [6, 34] to portray sequential features (*e.g.,* packet length sequence). While these methods are less suitable for detecting unknown attacks.

**NIDS with Unknown Attacks Detection.** These methods mainly involve three types of technologies: unsupervised, semi-supervised, and zero-shot learning. (i) *Unsupervised learning methods* such as clustering algorithms (*e.g.,* K-means [19], DBSCAN [9], and CSPA [54]) have been applied to identify outliers in network traffic. They are also known as "zero-positive" learning [7, 18] due to solely using benign samples for training. (ii) *Semi-supervised learning methods* such as Cls-Anomaly [65], FARE [31], and SENC [42] are usually composed of unsupervised and supervised learning. (iii) *Zero-shot learning methods* (ZSL) have been used to classify unknown classes in NIDS [44]. With the non-incremental learnability, and the need for rich "side information" to construct the feature mapping, ZSL methods are not suitable for our problem. Overall, their focus is different from ours, Trident devotes to fine-grained unknown class detection and ever-changing traffic adaption in an incremental manner.

**Some Recent Advances for NIDS.** Security communities propose a series of advanced research directions including: solutions based on programmable switches [4, 25, 37, 61, 69] to adapt to high-speed bandwidth. Leveraging formal verification to analyze the security of NIDS [58, 59, 72]. And some research devoted the automated characterization [20] and interpretability for NIDS [18, 40].

## 10 CONCLUSION

This paper presents Trident, a fine-grained traffic analysis framework towards identifying both known/unknown attack types, as well as adapting to variable traffic in an incremental manner. Based on our proposed framework, we implement three model architectures and extensively evaluate them on four public datasets. Moreover, we produce a series of experiments for Trident in terms of supervised, binary classification AD, and multi-classification AD. The results demonstrate the effectiveness and robustness of Trident outperforming existing SOTA methods.

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

# APPENDIX

# A  ADDITIONAL DETAILS OF TSIEVE

## A.1  Additional Details for AutoEncoder Architecture

For AutoEncoder, the 106-d features are displayed in Table 4.

**Table 4: The 106-d feature set. "TTL": "time to live".**

| Feature | | | | Dim |
|---|---|---|---|---|
| Protocol | | | | 1 |
| **Direction** | **Sum** | | | |
| | Frame | IP | TCP | |
| Forward | Packet_num, Duration | Flags_df, | ACK, URG, | 33 |
| Backward | | Flags_mf, | PUSH,RESET, | |
| Bi-dir | | Frag_offset | SYN, FIN | |
| **Direction** | **Statistic** | | | |
| | Max | Min | Mean | Std | |
| Forward | Total Length, Time_delta, | | | 72 |
| Backward | Calculated Window Size, | | | |
| Bi-dir | Scale_factor, Window, TTL | | | |

## A.2  Additional Details for RNN Architecture

The Recurrent Neural Network (RNN) is a natural generalization of feedforward neural networks to sequences. Given a sequence of inputs $(x_1, x_2, \cdots, x_T)$, a standard RNN computes a sequence of outputs $(y_1, y_2, \cdots, y_T)$ by iterating the following equation:

$$h_t = \text{sigmoid}(W^{\text{hx}} x_t + W^{\text{hh}} h_{t-1}) \quad (3)$$

$$y_t = W^{\text{yh}} h_t \quad (4)$$

where $W^{\text{hx}}$, $W^{\text{hh}}$, and $W^{\text{yh}}$ are the weight matrices of input-hidden, hidden-hidden, and hidden-output, respectively. The goal of the LSTM is to estimate the conditional probability $p(y_1, y_2, \cdots, y_T \mid x_1, x_2, \cdots, x_T)$, where $(x_1, x_2, \cdots, x_T)$ is an input sequence and $(y_1, y_2, \cdots, y_T)$ is its corresponding output sequence with the same length. The LSTM computes this conditional probability by first obtaining the fixed dimensional representation $v$ of the input sequence $(x_1, x_2, \cdots, x_T)$ given by the last hidden state of the LSTM, and then computing the probability of $(y_1, y_2, \cdots, y_T)$ with a standard LSTM-LM formulation whose initial hidden state is set to the representation $v$ of $(x_1, x_2, \cdots, x_T)$:

$$p(y_1, y_2, \cdots, y_T \mid x_1, x_2, \cdots, x_T) = \prod_{t=1}^{T} p(y_t \mid v, y_1, y_2, \cdots, y_{t-1}) \quad (5)$$

Notably, as detailed in Section A.2, different from the above LSTM model, we only use the structure of sequence-to-sequence RNN model to construct the hidden layer relationship of the inputs itself, that is, the correct outputs and inputs are consistent. In this way, we employ the MSE as the loss function of the training process.

## A.3  Additional Details for GNN Architecture

As shown in Fig. 6 and detailed in Section 4.3, the GNN architecture contains Graph Pooling Layer, Graph Unpooling Layer, and GCN Layer.

**Graph Pooling Layer.** The Graph Pooling (gPool) layer is used to enable down-sampling on graph data. In this layer, a subset of nodes is adaptively selected to form a new but smaller graph. To this end, a trainable projection vector p is employed. By projecting all node features to 1D, $k$-max pooling for node selection can be performed. Since the selection is based on 1D footprint of each node, the connectivity in the new graph is consistent across nodes. Given a node $i$ with its feature vector $X_i$, the scalar projection of $X_i$ on p is $y_i = X_i \text{p}/\|\text{p}\|$. Among them, $y_i$ measures how much information of node $i$ can be retained when projected onto the direction of p. Specifically, the graph can be represented by two matrices; those are the adjacency matrix $A \in \mathbb{R}^{N \times N}$ and the feature matrix $X \in \mathbb{R}^{N \times C}$. Each non-zero entry in the adjacency matrix $A$ represents an edge between two nodes in the graph. Each row vector $X_i$ in the feature matrix $X$ denotes the feature vector of node $i$ in the graph. The layer-wise propagation rule of the graph pooling layer $\ell$ is:

$$y = X^{\ell} \text{p}^{\ell} / \|\text{p}\| \quad (6)$$

$$\text{idx} = \text{rank}(y, k) \quad (7)$$

$$\tilde{y} = \text{y}(\text{idx}) \quad (8)$$

$$\tilde{X}^{\ell} = X^{\ell}(\text{idx}, :) \quad (9)$$

$$A^{\ell+1} = A^{\ell}(\text{idx}, \text{idx}) \quad (10)$$

$$X^{\ell+1} = \tilde{X}^{\ell} \odot (\tilde{y} 1_C^T) \quad (11)$$

where $k$ is the number of nodes selected in the new graph. $\text{rank}(y, k)$ is the operation of node ranking, which returns indices of the $k$-largest values in $y$. The idx returned by $\text{rank}(y, k)$ contains the indices of nodes selected for the new graph. $A^{\ell}(\text{idx}, \text{idx})$ and $X^{\ell}(\text{idx}, :)$ perform the row and/or column extraction to form the adjacency matrix and the feature matrix for the new graph. $\text{y}(\text{idx})$ extracts values in $y$ with indices idx followed by a sigmoid operation. $1_C \in \mathbb{R}^C$ is a vector of size $C$ with all components being 1, and $\odot$ represents the element-wise matrix multiplication.

**Graph Unpooling Layer.** To enable up-sampling operations on graph data, the graph unpooling (gUnpool) layer, which performs the inverse operation of the gPool layer and restores the graph into its original structure. To achieve this, we record the locations of nodes selected in the corresponding gPool layer and use this information to place nodes back to their original positions in the graph. Formally, we propose the layer-wise propagation rule of graph unpooling layer as:

$$X^{\ell+1} = \text{distribute}(0_{N \times C}, X^{\ell}, \text{idx}) \quad (12)$$

where $\text{idx} \in \mathbb{Z}^{*k}$ contains indices of selected nodes in the corresponding gPool layer that reduces the graph size from $N$ nodes to $k$ nodes. $X^{\ell} \in \mathbb{R}^{k \times C}$ are the feature matrix of the current graph, and $0_{N \times C}$ are the initially empty feature matrix for the new graph. $\text{distribute}(0_{N \times C}, X^{\ell}, \text{idx})$ is the operation that distributes row vectors in $X^{\ell}$ into $0_{N \times C}$ feature matrix according to their corresponding indices stored in idx). In $X^{\ell+1}$, row vectors with indices in idx) are updated by row vectors in $X^{\ell}$, while other row vectors remain zero.

**GCN Layer.** Notably, there is a GCN layer before each gPool layer, thereby enabling gPool layers to capture the topological information in graphs implicitly. Before the processing of the GCN layer, the $k$-th graph power $\mathbb{G}^k$ to increase the graph connectivity. We employ $k = 2$ since there is a GCN layer before each gPool layer to aggregate information from its first-order neighboring nodes.

**Algorithm 2** Incrementally Add Learner

---

**Require:** The learner list $L_p$, the threshold list $L_t$, the new class label set $\mathcal{Y}$ and the corresponding sample set $X$

**Ensure:** The updated learner list and the updated threshold list

1: **for** $y \in \mathcal{Y}$ **do**
2:     Select subset $b_i$ s.t. $(x, y) \in b_i \in X$
3:     Construct learner $p_i$ based on $b_i$
4:     $t_i \leftarrow Outlier\ Threshold(p_i, b_i)$
5:     Apeend $t_i$ to $L_t$
6:     Apeend $p_i$ to $L_p$
7: **end for**
8: **return** $L_p, L_t$

---

Formally, the equation is delivered:

$$A^2 = A^\ell A^\ell, \ A^{\ell+1} = A^2(\text{idx}, \text{idx}) \tag{13}$$

where $A^2 \in \mathbb{R}^{N \times N}$ is the 2-th graph power. Now, the graph sampling is performed on the augmented graph with better connectivity. In the GCNs, the layer-wise forward-propagation operation is defined as:

$$X_{\ell+1} = \sigma(\tilde{D}^{-\frac{1}{2}} \tilde{A} \tilde{D}^{\frac{1}{2}} X_\ell W_\ell) \tag{14}$$

where $\tilde{A} = \tilde{A} + 2I$ is used to add self-loops in the input adjacency matrix $A$, $X_\ell$ is the feature matrix of layer $\ell$. The GCN layer uses the diagonal node degree matrix $\tilde{D}$ to normalize $\tilde{A}$. $W_\ell$ is a trainable weight matrix that applies a linear transformation to feature vectors.

## A.4 Additional Details of Class-Level Incremental Update

The pseudocode for class-level incremental update is described in Algorithm 2.

## B ADDITIONAL DETAILS OF TSCISSORS

The Extreme Value Theory (EVT) [28] can be used to estimate the outlier bounds. Specifically, the form of Extreme Value Distributions (EVD) is presented as follows.

$$G_\gamma : x \mapsto \exp(-(1 + \gamma x)^{-\frac{1}{\gamma}}), \gamma \in \mathbb{R}, 1 + \gamma x > 0 \tag{15}$$

where $\gamma$ denotes the extreme value index. An elegant property of EVT is that the distribution of the extreme values is not dependent on the distribution of the data. In other words, these extreme events have the same kind of distribution, regardless of the original one, such as Frechet, Gamma, and Uniform distributions [51].

For a trained learner, we can obtain the reconstruction loss list for its corresponding data subset. The upper quantile of the list is used as the initialization threshold $t$. Then, according to Pickands-Balkema-de Haan theorem (also called second theorem in EVT) [1, 22], the extrema of cumulative distribution function $F$ converge in distribution to $G_\gamma$, if and only if a function $\sigma$ exists, i.e.,

$$\bar{F}_t(x) = \mathbb{P}(X - t > x | X > t) \underset{t \to \tau}{\sim} \left(1 + \frac{\gamma x}{\sigma(t)}\right)^{-\frac{1}{\gamma}} \tag{16}$$

It means that $X - t$ (excess over threshold) tends to follow a Generalized Pareto Distribution (GPD)[5] with parameters $\gamma, \sigma$. Once we

---

[5]The location $\mu$, the third parameter of GPD, is null in our case.

get estimates $\hat{\gamma}, \hat{\sigma}$, the Peaks-Over-Threshold (POT) approach could be used to calculate the threshold as follows

$$\mathcal{T} \simeq t + \frac{\hat{\sigma}}{\hat{\gamma}} \left( \left(\frac{qn}{N_p}\right)^{-\hat{\gamma}} - 1 \right) \tag{17}$$

where $q$ denotes the risk parameter, $n$ refer to data size, and $N_p$ represents the number of peaks.

In order to estimate $\hat{\gamma}, \hat{\sigma}$, the maximum likelihood estimation is considered an efficient method, and its goal is to maximizing (after logarithmic operation):

$$\log \mathcal{L}(\gamma, \sigma) = -N_p \log \sigma - \left(1 + \frac{1}{\gamma}\right) \sum_{i=1}^{N_p} \log\left(1 + \frac{\gamma}{\sigma}\mathcal{P}_i\right) \tag{18}$$

where $\mathcal{P} = \{l - t \text{ s.t. } l > t \text{ and } l \in L\}$ and $L$ denotes the loss list. According to [16, 51], Grimshaw's proposal could be used to reduce the two variables optimization problem to a single variable equation. Specifically, if we get a solution $(\gamma^*, \sigma^*)$, the variable $x^* = \gamma^*/\sigma^*$ is solution of the scalar equation $u(x)v(x) = 1$ where:

$$u(x) = \frac{1}{N_p} \sum_{i=1}^{N_p} \frac{1}{1 + x\mathcal{P}_i} \quad v(x) = 1 + \frac{1}{N_p} \sum_{i=1}^{N_p} \log(1 + x\mathcal{P}_i) \tag{19}$$

That is to say, by finding a solution $x^*$ of this equation, we can retrieve $\gamma^* = v(x^*) - 1$ and $\sigma^* = \gamma^*/x^*$. Finally, the estimates $\gamma^*$ and $\sigma^*$ can be used to calculate the threshold reference Eq. (17).

## C ADDITIONAL DETAILS FOR EVALUATION

We introduce and summarize the details for a series of baselines here (refer to § 7.1).

**(i) Supervised Method.**

- **nPrint** [20] is a tool that generates a unified packet representation and then leverages AutoML to fit the tabular data.
- **FlowPic** [49] processes packet length and timestamp fields and converts them into pictures and uses Convolutional Neural Networks (CNNs) to identify traffic.
- **FlowLens** [4] calculates statistical histograms of packet size distribution and adopts machine learning models (e.g., XGBoost) to perform classification.
- **ERNN** [70] integrates the finite state automaton inside the RNN unit to cope with network-induced phenomena such as packet loss, retransmission, and out-of-order.
- **ET-BERT** [32] handles the raw packets in hexadecimal and deploys a pre-trained transformer to represent and learn the contextualized datagram-level information.

**(ii) Binary Classification Anomaly Detection.**

- **Kitsune** [41] calculates a series of statistical features and designs an ensemble of AutoEncoders to detection intrusions.
- **Whisper** [13] expresses traffic as frequency domain information through the fast Fourier transform and then performs robust identification.
- **DeepLog** [8] utilizes Long Short-Term Memory (LSTM) [5] to model the system log and detects the anomalies in sequences.
- **HyperVision** [14] is an unsupervised malicious traffic detection system that could capture flow interaction patterns represented by the graph's structural features.

- **Diff-RF** [38] takes into the frequencies of visits in the leaves on the isolated forest [36] basis to detect point-by-point and collective anomalies.

**(iii) Multi-Classification Anomaly Detection.**

- **K-means** [19] and **DBSCAN** [9] are typical density-based and distance-based unsupervised clustering algorithms respectively.
- **FlowPrint** [56] is a semi-supervised approach for fingerprinting mobile apps from encrypted network traffic.

- **Cls-Anomaly** [65] employs Conditioned Variational AutoEncoder and extreme value theory to devote multi-classification for known attacks.
- **SENC** [42] completes the semi-supervised classification based on isolation forest, yet it assumes only to emerge one unknown class at one time.
- **FARE** [31] is a semi-supervised clustering method for classification under low-quality labels. Note that it needs to specify the number of classes for FARE, we set it as ground truth.

