# OpenReview forum: "Trident: A Universal Framework for Fine-Grained and Class-Incremental Unknown Traffic Detection"
_ACM.org/TheWebConf/2024/Conference — TheWebConf24 Oral_

### Official Review · Reviewer_JuP2 · 2023-11-04

**Novelty:** 4
**Technical Quality:** 5

**Review:**

Dear authors, thanks for submitting your work to WWW 2024. Detecting unknown attack traffic is a hard yet intriguing problem, and differentiating known attacks from unknown attacks definitely worths more attention from the research community. With that being said, I do feel this paper needs major improvement on both motivation and technical solutions. Please see details below:

1. It seems the motivation of the paper is to do "continuous integration" as new types of attacks emerge, without having to retrain models for known attacks. In order to distinguish different classes of unknown attacks, an ensemble clustering algorithm is used. My first question is, why not just use this clustering technique directly for multi-class anomaly detection? It seems the only issue is how to decide the number of clusters, but this doesn't seem to be a solvable problem anyway, even if you only use it for labeling. Concretely, why can't we do the following: (1) set up some threshold for certain existing multi-class detector (e.g. K-means), to distinguish unknown cases from known categories. (2) Once we have quite a few unknown cases in the "buffer", retrain the detector to take new data points into account. The paper in its current form does not convince me that a series of one-class learners are necessary. It seems to me that the most important problem---how to determine classes for unknown cases---and its relation with model training/inference is not well explained. The suggestion is to consider such "strawman solutions" and state why they do not work in your case.
2. What is the purpose of having three ML models (AE, RNN, GNN) with similar performance in the paper? If one model has superior performance compared with the other two, I think this strategy is acceptable. But now it looks they are basically on the same level, with some up-and-downs across different datasets, so I'm confused what is the key insights you want to show here. In the evaluation, it appears the proposed models have similar performance with existing ones, which means accuracy/precision improvement does not seem to be the primary contribution of the paper anyway. The concrete suggestion is to cut ML model discussion short if that's not important, other wise clarify why this choice matters.
3. How is "incremental update" and "unknown classes" actually evaluated? There is a lot of ambiguity in the paper---e.g. K-means is used for class-increments evaluation, where multiple known/unknown classes exist, but it is also mentioned in the paper that K-means is  "zero positive training" so cannot distinguish classes. The concrete suggestion is to clarify on these missing pieces.
4. Writing could be improved overall. For example "The advantage of the preset method is that it is convenient and prone to implement"

I'd appreciate any revision/response from the authors w.r.t the aforementioned problems going forward.

**Questions:**

The most important question, as I mentioned in the review, is I don't understand why a series of one-class learners are needed for handling unknown classes of attacks. Why not just design a simple threshold to warn against drifts/outliers/new categories, and retrain the model in time to align with the newest data points?
Some secondary questions include why discussing three ML models with similar performance in the paper, and how things like "incremental update" is evaluated.

**Reviewer Confidence:**

3: The reviewer is confident but not certain that the evaluation is correct

**Scope:**

3: The work is somewhat relevant to the Web and to the track, and is of narrow interest to a sub-community

---

### Official Review · Reviewer_24Ma · 2023-11-21

**Novelty:** 4
**Technical Quality:** 4

**Review:**

This paper proposes a universal framework (Trident) for fine-grained unknown encrypted traffic detection. It consists of three main modules, i.e., tSieve, tScissors, and tMagnifier are used for profilingtraffic, determining outlier thresholds, and clustering respectively. Experimental results demonstrate that Trident outperforms 16 state-of-the-art (SOTA) methods

Strengths:

1.The method is novel that includes three tightly coupled components (tSieve, tScis-sors, tMagnifier) for Trident

2.The experiments are extensive.

Weaknesses:

I think that the methods for compare may not be state-of-the-art, I recommend considering a more advanced approach, such as the FLDetector in [1], for a comprehensive evaluation.

[1] Zaixi Zhang, Xiaoyu Cao, Jinyuan Jia, and Neil Zhenqiang Gong. Fldetector: Defending federated learning against model poisoning attacks via detecting malicious clients. In Proceedings of the 28th ACM SIGKDD Conference on Knowledge Discovery and Data Mining, pages 2545–2555, 2022.

**Questions:**

See weaknesses.

**Reviewer Confidence:**

3: The reviewer is confident but not certain that the evaluation is correct

**Scope:**

4: The work is relevant to the Web and to the track, and is of broad interest to the community

---

### Official Review · Reviewer_SE4a · 2023-11-21

**Novelty:** 5
**Technical Quality:** 6

**Review:**

The paper develops a NIDS based on a series of 1-class models that recognize specific types of attacks, plus a mechanism to potentially learn from outliers new types of attacks.

Now, the use of multiple models adapted to specific pre-defined classes of attacks is not novel as a concept (see https://ieeexplore.ieee.org/abstract/document/4627082?casa_token=S8RF-s_lMoEAAAAA:HqdtdFo3b5CSoWDIl4Si1iugOXEqdVDMPQIBOpXFaCx3-41OI6aKBxxo0z13uHqVhZFbL-UC or https://www.sciencedirect.com/science/article/pii/S1566253506000765?casa_token=jjnpZQzvzCsAAAAA:YxwqYCnmy36P38TiwY-TUcAzV7GoIHsU0gwbPI_xXyadTcp8F6ObPMG-g3crkUR9yvGkIlQSIw), so the novelty should lie in the mechanism to automatically learn new types of attacks from the outliers.

The authors disregard payload contents due to the abundance of encrypted packets in today's networks. While certainly true, it should be noted that this is actually a step backward wrt detectors capable to ALSO analyze payloads.

The experimental section is thorough and I appreciated it, however it lacks the evaluation of the (most significant) improvement brought forth by the paper, the mechanism to automatically label "novel attacks". I would have expected, besides using internal metrics, that the authors would also externally validate for consistency, at least with a sampling approach, the learned classes. This was not done by the authors.

**Questions:**

None

**Ethics Review Description:**

-

**Reviewer Confidence:**

4: The reviewer is certain that the evaluation is correct and very familiar with the relevant literature

**Scope:**

4: The work is relevant to the Web and to the track, and is of broad interest to the community

---

### Official Review · Reviewer_QtTp · 2023-11-22

**Novelty:** 4
**Technical Quality:** 4

**Review:**

This paper proposes a new framework called Trident to detect both known and unknown attacks in the context of concept drift. Trident consists of three stages. In the first stage, AutoEncoders, recurrent neural networks (RNN), or bipartite GNNs are used to train one-class models for each known traffic type. These models can be updated incrementally at either the sample or class level. In the second stage, these models are used to identify outliers, where the thresholds can be estimated based on the extreme value theory. In the third stage, labels are assigned based on known attacks and unknown classes are identified through clustering algorithms. Four existing datasets are used for performance comparison against various other models. The results show that even with concept drifts, the proposed technique can still achieve high detection accuracy. The update times have been shown to be small at the millisecond level.

Strengths:
+ The paper is well written.
+ The proposed method tackles the problems of unknown attack detection in the context of concept drift, for which research still lacks.
+ The work used multiple datasets in its performance evaluation and compared the performances against various existing anomaly detection methods. The design of the experiments is thorough.

Weaknesses:
- The proposed framework requires a separate one-class model to be trained for each traffic type. For some complex computer networks (e.g., large ISPs), this may be computationally expensive.
- Some of the performance metrices used could have been explained more clearly.

**Questions:**

* In Section 7.4, AMI was used for performance comparison. However, it's unclear how it was calculated. This makes it hard to interpret the results in Figure 10.

* It's understandable that using per-traffic-type traffic classification models could lead to better detection accuracy, but they also lead to increased model complexity and operational overhead. The datasets used by this work may not be representative of the complex traffic types seen in large networks.

**Reviewer Confidence:**

2: The reviewer is willing to defend the evaluation, but it is likely that the reviewer did not understand parts of the paper

**Scope:**

2: The connection to the Web is incidental, e.g., use of Web data or API

---

### Official Review · Reviewer_35aH · 2023-11-30

**Novelty:** 4
**Technical Quality:** 4

**Review:**

This paper proposes a framework for network traffic analysis that is able to detect novel types of traffic (never observed before) and automatically create new classes accordingly.

Pros:
- The proposed framework is interesting and deals with an important and significant topic.
- Although a major proofreading is needed, the overall presentation quality is sufficient.
- The authors evaluated the proposed framework under multiple dimensions, considering also concept drift and overhead evaluation, very often neglected by previous work.

Cons:
- According to the adversary model and assumptions stated in Section2, it seems that the main goal of this paper is detecting and classifying new attacks (never seen before by the model). However, it is unclear how the proposed framework can autonomously decide whether a new detected class of traffic is malicious or legitimate.
- 3 out of 4 datasets used in this study are quite old (from 2016 to 2018) and possibly no longer representative of the current network traffic.
- The data used for concept drift evaluation (Section 7.5) are too close to each other (only one year). Therefore, the time bias evaluation is quite weak.
- The technical depth is not sufficient. For example, the exact composition of the databases used for evaluation are not reported (number of classes, number of instances per class, etc..). This makes it impossible to properly assess the soundness and meaningfulness of the evaluation experiments. In addition, the discussion in section 7.2, 7.3, and 7.4 is poor and lack important details.
- It is unclear how the authors selected the datasets. Some of them, such as USTC-TFC2016, include malicious and legitimate traffic, while others contain legitimate traffic only. In addition, why do they include TOR traffic?
- The related work section fails in drawing the state-of-the-art in anomaly-based NIDS. In addition, the authors do not sufficiently discuss how their solution improves it.

Other minors:
- A major proofreading is needed to fix several typos, such as (i) line 215: “testset include”; (ii) line 847: “refer to he extracted”; and incomplete sentences, such as in line 209.
- In line 597, the authors describe the existing solution instead of the dataset.
- in Table 1, first column (USTC/ACC), the value 95.65 (line 588) should not be bold (it is not the best accuracy for that experiment)
- Figure 1 is almost useless (at least the bottom part). In fact, tMagnifier, tSieve, tScissors are represented with icons that do not have a clear meaning. As a result, the figure is unintelligible by itself.

**Questions:**

-  when the model detects a new class, how can it know if the new class is a novel attack or a new class of benign traffic?
- can you give an intuition of how your framework can be used in a real-world scenario?

**Reviewer Confidence:**

3: The reviewer is confident but not certain that the evaluation is correct

**Scope:**

4: The work is relevant to the Web and to the track, and is of broad interest to the community

---

### Decision · Program_Chairs · 2024-01-22

**Decision:**

Accept (Oral)

**Comment:**

The paper is a sapient mix of technical ingenuity and extensive experimental support. The contribution is poised to improve the field of anomaly-based network intrusion detection systems (NIDSs). The reviewers, though at different degrees, agree on recognizing the value of the paper. The vivid exchange between reviewers and authors has clarified some of the issues related to the paper (though some minor issues persist).
 Among the top 2 papers in my pile.

 ---